# Prevalence and Risk Factors for Positive Nasal Methicillin-Resistant *Staphylococcus aureus* Carriage among Orthopedic Patients in Korea

**DOI:** 10.3390/jcm8050631

**Published:** 2019-05-08

**Authors:** Sung-Woo Choi, Jae Chul Lee, Jahyung Kim, Ji Eun Kim, Min Jung Baek, Se Yoon Park, Suyeon Park, Byung-Joon Shin

**Affiliations:** 1Department of Orthopedic Surgery, Soonchunhyang University Hospital, College of Medicine, Seoul 04401, Korea; jlee@schmc.ac.kr (J.C.L.); t0152@schmc.ac.kr (J.K.); 2Department of Laboratory Medicine, Soonchunhyang University College of Medicine, Seoul 04401, Korea; jkim@schmc.ac.kr; 3Department of Obstetrics and Gynecology, Bundang CHA Hospital, Seongnam 13496, Korea; goodgood75@naver.com; 4Division of Infectious Diseases, Department of Internal Medicine, Soonchunhyang University Hospital, College of Medicine, Seoul 04401, Korea; sypark@schmc.ac.kr; 5Department of Biostatistics, Soonchunhyang University College of Medicine, Seoul 04401, Korea; sue3517@schmc.ac.kr

**Keywords:** Methicillin-resistant *Staphylococcus aureus* (MRSA), nasal screening test, infection, prevalence, risk factors

## Abstract

Methicillin-resistant *Staphylococcus aureus* (MRSA) causes purulent skin and soft tissue infections as well as other life-threatening diseases. Recent guidelines recommend screening for MRSA at the time of admission. However, few studies have been conducted to determine the prevalence and risk factors for MRSA colonization. A prospective data collection and retrospective analysis was performed. MRSA screening tests were performed using nasal swabs in patients enrolled between January 2017 and July 2018. Demographic data, socio-economic data, medical comorbidities, and other risk factors for MRSA carriage were evaluated among 1577 patients enrolled in the study. The prevalence of MRSA nasal carriage was 7.2%. Univariate regression analysis showed that colonization with MRSA at the time of hospital admission was significantly related to patient age, body mass index, smoking, alcohol, trauma, recent antibiotic use, and route of hospital admission. Multiple logistic regression analysis for the risk factors for positive MRSA nasal carriage showed that being under- or overweight, trauma diagnosis, antibiotic use one month prior to admission, and admission through an emergency department were related to MRSA colonization. This study highlights the importance of a preoperative screening test for patients scheduled to undergo surgery involving implant insertion, particularly those at risk for MRSA.

## 1. Introduction

The emergence of methicillin-resistant *Staphylococcus aureus* (MRSA) since the early 1960s and its spread throughout hospitals and communities decades later have complicated antibiotic therapy [1,2,3,4,5]. The rate of methicillin resistance is higher in the orthopedics department, where artificial implantation is more common than in other medical specialties [6]. If the infection is caused by MRSA, either due to antibiotic resistance or biofilm formation, the treatment of an orthopedic infection becomes difficult and may lead to a higher economic burden [7]. Therefore, early detection and prevention of MRSA are important, particularly among orthopedic patients.

The recent guidelines published by the Society for Health Care Epidemiology for America recommend surveillance of cultures at the time of hospital admission for patients scheduled to undergo high-risk surgeries, including some orthopedic and cardiothoracic procedures [8,9]. The Korea Center for Disease Control and Prevention suggests a screening test only for patients who are admitted to high-risk departments such as the intensive care unit, the hemato-oncology department, and the organ transplantation department; patients with a prior diagnosis of MRSA; and patients transferred from a nursing facility [10]. The recommended screening method for MRSA is nasal sampling because the anterior nasal cavity is one of the preferred carrier sites of this bacterium and because the frequency of skin colonization depends on nasal carriage [11,12]. In addition, Yano et al. [13] reported that patients with positive preoperative nasal cultures for MRSA had a higher occurrence of surgical site infection with MRSA after orthopedic surgery.

However, a limited number of studies have been conducted to determine the prevalence and risk factors for colonization at the time of admission in orthopedic patients [14,15,16,17,18], especially in Korea. The prevalence of MRSA is extremely high in Korea. The Asian Network for Surveillance of Resistant Pathogens (ANSORP) study, which included seven hospitals in Korea, showed an average MRSA prevalence of 77.6% for nosocomial *S. aureus* isolates [19]. The recent report of the Regional Resistance Surveillance (RRS) program showed that 73% of the clinical *S. aureus* isolates from two hospitals in Korea were MRSA [20]. Korea has the highest MRSA prevalence among the 12 surveillance countries in the RRS program [20,21].

The present study primarily aimed to assess the prevalence of MRSA carriage in the orthopedic department. The secondary objective was to identify potential risk factors for MRSA colonization with respect to the demographic and medical data.

## 2. Material and Methods

The study involved prospective data collection and retrospective data analysis and was conducted between January 2017 and July 2018 at the orthopedic surgery department of Soonchunhyang University Hospital and approved by the Institutional Review Board (IRB No.: SCHUH 2018-11-021).

### 2.1. Inclusion and Exclusion Criteria

The patients admitted to the orthopedic surgery department of Soonchunhyang University Hospital, Korea were enrolled in the study based on the following inclusion criteria: patients aged 18 to 90 years were included in the study and patients with a current infection were excluded.

### 2.2. Sampling

The patients, who were admitted to the hospital during the study period and satisfied the inclusion criteria, had nasal swabs collected from them. All sampling was performed by well-trained orthopedic nurses. Nasal swabs were obtained within a day of admission. A sterile transport swab (COPAN, Brescia, Italy) was rotated in the anterior nasal cavity of the patients by the nurses. The swab was transported at room temperature and each swab was processed as described here within four hours of collection. Direct culture was performed onto chromID MRSA agar (bioMérieux, Nürtingen, Germany) and was examined at 24 and 48 hours after incubation. A positive culture was defined as growth with morphological features comparable to MRSA. This was confirmed by the coagulase test using a commercial latex agglutination kit (Pastorex Staph Plus, Bio Rad Laboratories, Hemel Hempstead, UK).

### 2.3. Data Collection

Data collection was coordinated by an orthopedic surgeon who completed standardized forms for each included patient. Overall data was recorded for the following variables: demographic characteristics, socio-economic characteristics, medical comorbidities, and other risk factors for MRSA carriage. Sex, age, body mass index (BMI), history of smoking and alcohol consumption, diagnosis of trauma or disease, and body part involved were analyzed as demographic data. BMI was divided into four groups (underweight (<18.5 kg/m^2^), normal weight (18.5–25 kg/m^2^), overweight (25–30 kg/m^2^), and obese (>30 kg/m^2^)) [22]. Smoking history was classified into three groups (never-smoker, ex-smoker, and current-smoker). An ex-smoker was defined as someone who had smoked more than 100 cigarettes in their lifetime but had not smoked in the last 28 days [23]. A current-smoker was defined as an adult who had smoked more than 100 cigarettes in his or her lifetime and was still smoking. History of alcohol consumption was divided into three groups (never-drinker, former-drinker, and current-drinker). A current-drinker was a person who consumed up to 12 drinks per year, while a former-drinker was a patient who had stopped drinking for >1 year [24]. The body part involved was divided into five categories (spine, knee and shoulder, hand and elbow, hip, and foot). Using the ICD-10-CM (International Classification of Diseases, Tenth Revision, Clinical Modification) codes, patients were divided into two groups (coded as trauma (S) and disease (M)). Education status was collected as socio-economic data [25,26,27]. Medical comorbidities included hypertension, diabetes mellitus, cardiovascular diseases, hepatic diseases, and dialysis. Other risk factors for MRSA carriage were recent hospitalization, recent antibiotics use, presence of a ureteral catheter, and type of hospital admission (i.e., whether emergency or outpatient department) (Table 1). Recent hospitalization and recent antibiotic use were defined as stationary hospitalization within 12 months prior to admission and antimicrobial therapy within one month before the screening test, respectively [28].

### 2.4. Statistical Analysis

Statistical analysis was performed using parametric or nonparametric tests, where appropriate. Logistic regression analysis was used to identify independent predictors of colonization with MRSA at the time of admission to the hospital as well as clinical conditions associated with the development of symptomatic MRSA infection. The association between the risk factors and MRSA colonization was analyzed using the chi-square test or Fisher’s exact test, as appropriate. Data were analyzed using SPSS Statistics (Statistical Package for the Social Sciences, version 25.0; 2017, IBM Corp. Armonk, NY, USA) and two-tailed *p*-values of ≤0.05 were considered as statistically significant.

## 3. Results

### 3.1. Overall Result

A total of 1577 patients were enrolled during the study period. Of these, 617 (39.1%) were women and 960 (60.9%) were men. The average patient age was 59.2 years; particularly, 288 patients (18.3%) were aged under 40 years, 776 (49.2%) were aged between 40 and 70 years, and 513 (32.5%) were aged older than 70 years. A total of 1332 patients (84.5%) were tested in the ward after admission, while 245 (15.5%) were tested in the outpatient department. Ultimately, 114 patients were found to be colonized with MRSA; thus, the prevalence of nasal MRSA carriage was 7.2%. Of the 114 colonized samples, 99 (86.8%) samples were collected after admission and 15 (13.2%) samples were collected in the outpatient department.

### 3.2. Risk Factors

#### 3.2.1. Univariate Logistic Regression Analysis

Selected variables of interest were subjected to analysis (Table 2 and Table 3). The results of the univariate logistic regression analysis showed that the colonization with MRSA at the time of hospital admission was significantly related to patient age, BMI, smoking, alcohol consumption, body part involved, diagnosis of trauma, type of hospital admission, and recent antibiotic use (Table 4). Patients aged 40 to 70 years and those older than 70 years had 0.237 (95% Confidence Interval (CI): 0.151–0.374; *p* < 0.001) and 0.292 (95% CI: 0.180–0.474; *p* < 0.001) times the risk of being a carrier of MRSA as compared to those aged under 40 years, respectively. Additionally, patients whose BMI was under 18.5 kg/m^2^, 25 to 30 kg/m^2^, and higher than 30 kg/m^2^ had 3.432 (95% CI: 1.308–9.000; *p* = 0.012), 0.290 (95% CI: 0.153–0.550; *p* < 0.01), 0.637 (95% CI: 0.238–1.702; *p* = 0.369) times the risk of being an MRSA carrier than those with a normal BMI. Current-smokers had an odds ratio of 4.204 (95% CI: 2.801–6.309; *p* < 0.01) and current-drinkers had an odds ratio of 4.204 (95% CI: 3.250–7.223; *p* < 0.001). Patients diagnosed with trauma had 3.401 (95% CI: 1.593–7.261; *p* < 0.01) times the risk of being an MRSA carrier than those diagnosed with disease. Furthermore, those who were recently treated with antibiotics tended to have an increased chance of being an MRSA carrier, with an odds ratio of 4.199 (95% CI: 2.831–6.228; *p* < 0.01). Lastly, patients admitted through the emergency department had 6.915 (95% CI: 2.863–16.702; *p* < 0.01) times the risk of being an MRSA carrier than those admitted through the outpatient department. Meanwhile, there was no strong association between MRSA carriage and other medical comorbidities, recent hospitalization, the presence of a urinary catheter, and socio-economic background.

#### 3.2.2. Multiple Logistic Regression Analysis

Table 4 shows the results of the multiple logistic regression analysis for the risk factors found statistically significant in the univariate logistic regression analysis. Being underweight, having trauma, recent antibiotic use, and admission through the emergency room were found to be risk factors. Patients with a BMI lower than 18.5 kg/m^2^ had a 2.026 times higher risk of nasal MRSA carriage than patients with normal BMI. Patients diagnosed with trauma had a 1.795 times higher rate of nasal MRSA carriage than those diagnosed with disease. In addition, patients with recent antibiotics use had a 1.946 times higher risk of MRSA carriage than patients without antibiotic use. Furthermore, patients admitted through the emergency department had a 3.998 times higher MRSA carriage rate than those admitted through the outpatient department.

## 4. Discussion

According to this study, the overall prevalence of nasal MRSA carriage is 7.2%. There is a huge difference in the epidemiology of MRSA worldwide. For example, the prevalence of community-associated MRSA infection in Japan, Germany, Turkey, Taiwan, and Malta was found to be 0.94% [29], 1.2% [17], 1.2% [30], 3.8% [31], and 8.81% [6] respectively. These differences can be attributed to variations in microbiological methods (sampling technique, culture sites, and method of MRSA identification), local infection control standards, and the local prevalence of MRSA. Gi et al. [32] reported that the methicillin resistance rate of staphylococcal isolates in Korea is slightly higher than that in other countries. Another study showed that the prevalence of MRSA is 73% among the nosocomial *S. aureus* isolates in Korea, which is higher than that in other Asian countries [21]. Therefore, we considered it worthwhile to study the prevalence and risk factors for MRSA colonization in the orthopedic field, where aseptic surgery is essential.

Several studies have evaluated the risk factors associated with MRSA carriage [33,34,35,36,37,38]. Of the many known risk factors, the present study identified being underweight, having a diagnosis of trauma, recent antibiotic use, and admission through the emergency department as significant risk factors for MRSA carriage. Contrary to some prior studies, the present study did not reveal age as a significant risk factor [35,36]. The majority of studies have reported that the elderly patients have a higher tendency of being an MRSA carrier. An increased burden of infection in the elderly is linked to age-related decline in immune function, malnutrition, and anatomical and physiological changes [35]. However, some studies have reported that MRSA colonization is not influenced by age [39], and others found a higher carriage rate in younger patients [36]. Therefore, the association between age and MRSA colonization remains controversial.

Our results showed that being underweight is a risk factor for MRSA colonization and being overweight decreases the colonization rate. However, obesity did not turn out to be a significant risk factor for MRSA colonization in this study. Neidhart et al. [17] reported a reduced risk for *S. aureus* carriage in obese (BMI ≥ 30.0 kg/m^2^) compared to overweight patients (BMI of 25.0 to 30 kg/m^2^). However, other studies reported the opposite result. Olsen et al. [40] found a significant positive correlation between BMI and MRSA carriage only in women, particularly among those aged 30–43 years. Campbell et al. [41] reported an increased risk of *S. aureus* colonization in patients with both obesity and asthma. The analysis of skin and soft tissue staphylococcal infections showed that obesity is related to the presence of methicillin resistance. Therefore, further studies are needed to prove a definite relation between BMI and MRSA colonization.

In this study, smoking was not found to be a statistically significant risk factor for MRSA carriage. The influence of smoking on the colonization of MRSA is still controversial. Some studies proposed that smokers have higher rates of MRSA colonization than nonsmokers, thus increasing their risk of serious and difficult-to-treat infections [42,43]. Ellisa et al. reported that cigarette smoke increases MRSA hydrophobicity, thus increasing MRSA adherence and invasion [43]. They also stated that cigarette smoking increases the MRSA expression of genes linked to cell surface changes. In contrast, the smoke itself seems to influence the *S. aureus* load as indicated by the reduced identification rate in the upper respiratory tract [44]. A possible explanation for this finding is the inherent toxicity of smoke. The capability of smoke to inhibit the growth capacity of Gram-positive bacteria, particularly *S. aureus*, has already been demonstrated [45]. However, whether smoking affects MRSA colonization remains unclear.

In our study, patients admitted through the emergency department and those diagnosed with trauma had 3.998 and 1.795 times higher risk of MRSA colonization, respectively. Most trauma patients tend to visit the emergency department, and these two factors were considered by us to be similar. Quach et al. [46] reported that a visit to the emergency department is associated with more than threefold increased risk of acute infection. Patients with morbid conditions are more likely to visit the emergency department than patients with less morbid conditions. In addition, admission through the emergency department may increase the chance of contact with these patients, which may be expected to increase the risk of acquiring an infection. Also, patterns of injury differ between different mechanisms of trauma [47] which may cause patients diagnosed with trauma to be more susceptible to MRSA. For example, certain mechanisms of trauma may result in skin injuries, which may serve as a portal for entry. Furthermore, overcrowding of the emergency department creates more opportunities for cross-transmission [48], and overworked medical staff may not follow preventive procedures and take inadequate precautions [49,50]. Our findings on the correlation between trauma, admission through the emergency department, and MRSA detection may be due to these factors.

This study proves the relationship between antibiotic use prior to admission and MRSA colonization. Muller et al. [28] reported individual exposure to fluoroquinolones and collective exposure to penicillin to be associated with MRSA isolation after adjustment for colonization pressure and other potential confounders. Tacconelli et al. [51] performed a meta-analysis of over 76 studies included 24,230 patients and found that the development of MRSA is significantly related to the length of antibiotic exposure. According to the Society for Healthcare Epidemiology of America guidelines for preventing nosocomial transmission of MRSA and vancomycin-resistant enterococci, the use of antimicrobial agents in US hospitals is commonly excessive or unnecessary [9]. This widespread use of antibiotics may create conditions in which resistant bacteria experience a competitive advantage.

There are some limitations to our study. First, only nasal swabs were performed by us as the screening test considering cost- and time-effectiveness. Some studies suggest that collecting samples from additional sites (throat, groin, and thorax) may contribute to a higher detection rate than using a single site [17]. The prevalence of MRSA colonization might have been higher if samples from additional sites were collected. However, the anterior nasal cavity is known to be the most common carrier site of MRSA [11,12]. Second, we focused mainly on the screening test but whether the result of the screening test actually contributed to the development of post-operative infection has not been investigated. Third, this is a single-institute and single-department study. Additional multi-center studies must be performed to determine the actual frequency and risk factors for MRSA in Korea.

However, the current study is valuable in that it is among the first to focus on the prevalence and risk factors for MRSA in the orthopedic patients in Korea. The importance of the nasal MRSA screening test will be further validated by analyzing the clinical outcomes of the patients with a positive swab.

## 5. Conclusions

The present study focused on the prevalence of MRSA colonization and associated risk factors among patients admitted to the orthopedic surgery department. Being underweight, having a trauma diagnosis, antibiotic use one month prior to admission, and admission through the emergency department are risk factors for MRSA infection. The relatively high prevalence of MRSA in this study highlights the importance of pre-operative screening tests for patients scheduled for surgery involving implant insertion, particularly those at risk for MRSA. Among high risk patients, elective surgery could be delayed until the confirmation of the MRSA screening test. Prophylactic treatment prior to surgery is recommended for patients with positive MRSA colonization.

## Figures and Tables

**Table 1 jcm-08-00631-t001:** Overview of collected patient data.

Patient Data	Type of Acquired Information
**Demographic**	Gender
	Age
	Body mass index (BMI)
	Smoking (current-smoker/ex-smoker/never-smoker)
	Regular consumption of alcohol (current-drinker/former-drinker/never-drinker)
**Socio-economic**	Status of education (under high school/high school graduate/denied or unanswered)
**Medical comorbidities**	Hypertension, diabetes mellitus, cardiovascular diseases, hepatic disease, dialysis
**Other risk factors**	Stationary hospitalization in last 12 months
	Antimicrobial therapy in the past month
	Presence of urethral catheter
	Route of admission (ER/OPD)
	Trauma/disease
	Body parts involved (spine/knee and shoulder/hand and elbow/hip/foot)

ER = Emergency Room, OPD = Outpatient department.

**Table 2 jcm-08-00631-t002:** Descriptive logistic regression analysis from data obtained at admission (demographics, socio-economic status).

	Number of Patients	MRSA (−) (1463 Patients)	MRSA (+) (114 Patients)	*p*-Value ^a^
	N = 1577 (%)	N (%)	N (%)	
Gender				0.359
Male	617 (39.1%)	577 (93.5%)	40 (6.5%)	
Female	960 (60.9%)	886 (92.3%)	74 (7.7%)	
Age group				<0.01
<40	288 (18.3%)	239 (82.9%)	49 (17.1%)	
40–70	776 (49.2%)	740 (95.4%)	36 (4.6%)	
≥70	513 (32.7%)	484 (94.3%)	29 (5.7%)	
BMI group				<0.01
Underweight (<18.5)	159 (10.1%)	116 (73.0%)	43 (27.0%)	
Normal (18.5–25)	735 (46.6%)	686 (93.3%)	49 (6.7%)	
Overweight (25–30)	566 (35.9%)	549 (97.0%)	17 (3.0%)	
Obese (≥30)	117 (7.4%)	112 (95.7%)	5 (4.3%)	
Smoking				<0.01
Never-smoker	1294 (82.1%)	1226 (94.7%)	68 (5.3%)	
Current-smoker	238 (15.1%)	193 (81.1%)	45 (18.9%)	
Ex-smoker	45 (2.9%)	44 (97.8%)	1 (2.2%)	
Alcohol				<0.01
Never-drinker	1292 (81.9%)	1228 (95.0%)	64 (5.0%)	
Current-drinker	248 (15.7%)	198 (79.8%)	50 (20.2%)	
Former-drinker	37 (2.3%)	37 (100.0%)	0 (0.0%)	
Status of education				0.372
Under high school	89 (5.6%)	82 (92.1%)	7 (7.9%)	
High school graduate	420 (26.6%)	374 (89.0%)	46 (11.0%)	
Denied, Unanswered	1068 (67.7%)	1007 (94.3%)	61 (5.7%)	

x% = Column percentage; N (xx%) = Number of patients (row percentage). OR = Odds ratio; CI = Confidence Interval; a = chi-square test. MRSA = Methicillin-resistant *Staphylococcus aureus*.

**Table 3 jcm-08-00631-t003:** Descriptive logistic regression analysis from data obtained at admission (medical comorbidities, other risk factors for MRSA).

	Number of Patients	MRSA (−) (1463 Patient)	MRSA (+) (114 Patient)	*p*-Value ^a^
	N = 1577 (%)	N (%)	N (%)	
Hypertension				0.059
No	919 (58.3%)	843 (91.7%)	76 (8.3%)	
Yes	658 (41.7%)	620 (94.2%)	38 (5.8%)	
Cardiovascular disease				0.510
No	1486 (94.2%)	1377 (92.7%)	109 (7.3%)	
Yes	91 (5.8%)	86 (94.5%)	5 (5.5%)	
Diabetes Mellitus				0.697
No	1236 (78.4%)	1145 (92.6%)	91 (7.4%)	
Yes	341 (21.6%)	318 (93.3%)	23 (6.7%)	
Hepatic disease				0.516
No	1502 (95.2%)	1392 (92.7%)	110 (7.3%)	
Yes	75 (4.8%)	71 (94.7%)	4 (5.3%)	
Dialysis				0.405 *
No	1565 (99.2%)	1451 (92.7%)	114 (7.3%)	
Yes	12 (0.8%)	12 (100.0%)	0 (0.0%)	
Recent hospitalization				0.951
No	1172 (74.3%)	1087 (92.7%)	85 (7.3%)	
Yes	405 (25.7%)	376 (92.8%)	29 (7.2%)	
Recent antibiotics				<0.01
No	1105 (70.1%)	1061 (96.0%)	44 (4.0%)	
Yes	472 (29.9%)	402 (85.2%)	70 (14.8%)	
Urinary catheter				0.985
No	1329 (84.3%)	1233 (92.8%)	96 (7.2%)	
Yes	248 (15.7%)	230 (92.7%)	18 (7.3%)	
Route of admission				<0.01
ER	227 (14.4%)	174 (76.7%)	53 (23.3%)	
OPD	1350 (85.6%)	1289 (95.5%)	61 (4.5%)	
Body part involved				0.343
Spine	567 (36.0%)	543 (95.8%)	24 (4.2%)	
Knee, shoulder	531 (33.7%)	506 (95.3%)	25 (4.7%)	
Hand, elbow	64 (4.1%)	64 (100.0%)	0 (0.0%)	
Hip	127 (8.1%)	119 (93.7%)	8 (6.3%)	
Foot	288 (18.1%)	231 (80.2%)	57 (19.8%)	
Disease/Trauma				<0.01
Trauma	560 (35.5%)	500 (89.3%)	60 (10.7%)	
Disease	1017 (64.5%)	963 (94.7%)	54 (5.3%)	

x% = Column percentage; N (xx%) = Number of patients (row percentage). OR = Odds ratio; CI = Confidence Interval. a = chi-square test or * Fisher’s exact test.

**Table 4 jcm-08-00631-t004:** Univariable and multivariable logistic regression analysis of the risk factors associated with MRSA colonization.

Variables	Univariable Analysis OR (95% CI)	*p*-Value	Multivariable Analysis Adjusted OR (95% CI)	*p*-Value
Age				
<40	1			
40–70	0.237 (0.151–0.374)	**<0.01 ****		
>70	0.292 (0.180–0.474)	**<0.01 ****		
BMI				
Normal weight	1		1	
Underweight	3.432 (1.308–9.000)	**0.012 ****	2.026 (1.115–3.680)	**0.020****
Overweight	0.290 (0.153–0.550)	**<0.01 ****	0.389 (0.219–0.689)	**<0.01 ****
Obese	0.637 (0.238–1.702)	0.369	0.637 (0.246–1.650)	0.354
Smoking				
Never-smoker	1			
Current-smoker	4.204 (2.801–6.309)	**<0.01 ****		
Ex-smoker	0.410 (0.056–3.019)	0.381		
Alcohol				
Never-drinker	1			
Current-drinker	4.204 (3.250–7.223)	**<0.01 ****		
Former-drinker	N/A			
Hypertension	0.680 (0.454–1.017)	0.061		
Code				
Disease	1		1	
Trauma	3.401 (1.593–7.261)	**<0.01 ****	1.795 (0.984–3.275)	**0.048 ****
Recent antibiotics use	4.119 (2.831–6.228)	**<0.01 ****	1.946 (1.178–3.217)	**<0.01 ****
Route of admission				
OPD	1		1	
ER	6.915 (2.863–16.702)	**<0.01 ****	3.998 (2.003–7.979)	**0.047 ****

**: *p* < 0.05.

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
