# Peer review of "Prevalence and Risk Factors for Positive Nasal Methicillin-Resistant Staphylococcus aureus Carriage among Orthopedic Patients in Korea"

_jcm, 2019, doi:10.3390/jcm8050631_

Reviewer 1 Report

The MRSA carriage is a well-known fact. The present study reconfirm the known data, and brings some specific associations on orthopedic patients.

There are many English spelling errors, "invented" words: see line 44 - portage; line 66 - nares; Passive voice should be considered throughout the manuscript (line 76). English grammar should be overall improved.

Lines 48-49 - there are many studies on MRSA carriage, including in orthopedics wards (a quick pubmed search shows this).

Lines 64-65 - were the patients supervised during self-collection of nasal swab? If not, this could be a severe bias.

Line 67 - "randomly used" - what does it mean? It is not clear.

Lines 69-71 - Colonies grown on selective chromogenic medium should be verified: are they indeed MR S. aureus? This would be a serious flaw of the study design. There are many limitations of chromogenic media (see the sensitivity, specificity, limitations in the product insert). Normally, the suspected colonies should be verified at least with coagulase test and cefoxitin screen test.

Figure1 is not suggestive. Some reports from some countries were selected and presented. There are many other reports from allover the world. Why were only these few ones presented? In the same idea, Table V could be extended (or removed and instead presented inline some risk factors related data).

Line 201 - Gram with capital G

Lines 204-216 - the discussion regarding trauma patients is illogical in the context of this study. Indeed, trauma patients may easier contact MRSA INFECTION, but the nose colonization is not related to the trauma (but rather with the close contact with ER / healthcare workers / other infected patients.

Lines 227-230 - Conversely, many other studies claim that nasal swabs are the ideal methods of MRSA detection, and axillar/groin swabs are unnecessary (cost and time consuming).

In my opinion, the article need major revisions and clarifications on the methodology. Otherwise, the interpretation and presentation of results is good. The conclusions support the presented data.

Author Response

Response to Reviewer 1 Comments

Comments and Suggestions for Authors

The MRSA carriage is a well-known fact. The present study reconfirm the known data, and brings some specific associations on orthopedic patients.

Point 1: There are many English spelling errors, "invented" words: see line 44 - portage; line 66 - nares;

Response 1: Thank you for your comment. We have revised the manuscript and corrected the spelling and grammar mistakes (portage à colonization, nares à nasal cavity).

Point 2: Passive voice should be considered throughout the manuscript (line 76).

Response 2: Thank you for your comment. We have changed the sentences from active voice to passive voice (line 96, 101).

Point 3: English grammar should be overall improved.

Response 3: Thank you for this suggestion. We have edited the document for grammar and readability.

Point 4: Lines 48-49 - there are many studies on MRSA carriage, including in orthopedics wards (a quick pubmed search shows this).

Response 4: Thank you for your insightful advice. We have presented the references in the above sentence (line 60). We typed “Orthopedic surgery MRSA nasal colonization” in PubMed and reviewed all 33 articles as well as 1 related article (See the references located at the end of this response letter). We found that not all articles focused on the prevalence and risk factors of MRSA nasal colonization in orthopedic patients. Some articles focused on the decolonization of MRSA [1,2] and others were molecular studies [3,4]. Some of the articles focused on MRSA colonization among orthopedic surgeons rather than patients [5,6]. Only 5 articles satisfied the criteria (Prevalence and risk factors of orthopedic patients) [7-11]. Moreover, the number of patients included in our study (1577 patients) turned out to be relatively large compared with the number in these studies. We suggest that studies on the prevalence and risk factors of MRSA nasal colonization among orthopedic patients are limited; therefore, our results are valuable. (We have added additional files regarding the description of the searched articles.)

Point 5: Lines 64-65 - were the patients supervised during self-collection of nasal swab? If not, this could be a severe bias.

Response 5: Thank you for your comment. Nasal swabs were collected by well-trained orthopedic nurses after admission and not self-collected. We have clarified our sampling method in the manuscript (line 81, 83).

Point 6: Line 67 - "randomly used" - what does it mean? It is not clear.

Response 6: We apologize for the mistake. The word ”randomly” has been deleted since all swabs were processed according to the same procedures as discussed in the manuscript (line 84).

Point 7: Lines 69-71 - Colonies grown on selective chromogenic medium should be verified: are they indeed MR S. aureus? This would be a serious flaw of the study design. There are many limitations of chromogenic media (see the sensitivity, specificity, limitations in the product insert). Normally, the suspected colonies should be verified at least with coagulase test and cefoxitin screen test.

Response 7: By the term “phenotypic features,” we implied those features that were observed by morphological examination. Further, the suspected colonies have been verified by the coagulase test, which was conducted using a commercial latex agglutination kit (Pastorex Staph Plus, Bio Rad Laboratories, Hemel Hempstead, UK) (line 87-88).

Point 8: Figure1 is not suggestive. Some reports from some countries were selected and presented. There are many other reports from all-over the world. Why were only these few ones presented?

Response 8: Thank you for your comment. As you mentioned, there are many studies performed in other countries on the prevalence of nasal MRSA. However, we compared our result with only a few of these studies, which could lead to selection bias. Therefore, we decided to remove the Figure 1 and retained only some of the listed data from other countries in the manuscript as an example.

Point 9: In the same idea, Table V could be extended (or removed and instead presented inline some risk factors related data).

Response 9: Thank you for your comment. As Table 1 lists some of the known risk factors, we have decided to remove Table 5 instead.

Point 10: Line 201 - Gram with capital G

Response 10: We have capitalized the word “gram” in the revised manuscript (line 225).

Point 11: Lines 204-216 - the discussion regarding trauma patients is illogical in the context of this study. Indeed, trauma patients may easier contact MRSA INFECTION, but the nose colonization is not related to the trauma (but rather with the close contact with ER / healthcare workers / other infected patients.

Response 11: Thank you for your detailed feedback. We believed that most of the trauma patients visited the emergency department and considered these two factors as similar. We added a comment addressing this in the manuscript (line 228-230) and included additional references regarding emergency department visit (line 232-234, 237-239).

Point 12: Lines 227-230 - Conversely, many other studies claim that nasal swabs are the ideal methods of MRSA detection, and axillar/groin swabs are unnecessary (cost and time consuming).

Response 12: Thank you for your comment. As you have mentioned, we performed only nasal screening as it was both cost- and time-effective. Therefore, we added our opinion in the manuscript (line 253).

In my opinion, the article need major revisions and clarifications on the methodology. Otherwise, the interpretation and presentation of results is good. The conclusions support the presented data.

Reference

1.        Bajolet, O.; Toussaint, E.; Diallo, S.; Vernet-Garnier, V.; Dehoux, E. [Is it possible to detect Staphylococcus aureus colonization or bacteriuria before orthopedic surgery hospitalization?]. Pathologie-biologie 2010, 58, 127-130.

2.        Chen, A.F.; Heyl, A.E.; Xu, P.Z.; Rao, N.; Klatt, B.A. Preoperative decolonization effective at reducing staphylococcal colonization in total joint arthroplasty patients. The Journal of arthroplasty 2013, 28, 18-20.

3.        Trouillet-Assant, S.; Valour, F.; Mouton, W.; Martins-Simoes, P.; Lustig, S.; Laurent, F.; Ferry, T. Methicillin-susceptible strains responsible for postoperative orthopedic infection are not selected by the use of cefazolin in prophylaxis. Diagnostic microbiology and infectious disease 2016, 84, 266-267.

4.        Post, V.; Harris, L.G.; Morgenstern, M.; Geoff Richards, R.; Sheppard, S.K.; Fintan Moriarty, T. Characterization of nasal methicillin-resistant Staphylococcus aureus isolated from international human and veterinary surgeons. Journal of medical microbiology 2017, 66, 360-370.

5.        Schwarzkopf, R.; Takemoto, R.C.; Immerman, I.; Slover, J.D.; Bosco, J.A. Prevalence of Staphylococcus aureus colonization in orthopaedic surgeons and their patients: a prospective cohort controlled study. The Journal of bone and joint surgery. American volume 2010, 92, 1815-1819.

6.        Morgenstern, M.; Erichsen, C.; Hackl, S.; Mily, J.; Militz, M.; Friederichs, J.; Hungerer, S.; Buhren, V.; Moriarty, T.F.; Post, V.; et al. Antibiotic Resistance of Commensal Staphylococcus aureus and Coagulase-Negative Staphylococci in an International Cohort of Surgeons: A Prospective Point-Prevalence Study. PLoS One 2016, 11, e0148437.

7.        Nelwan, E.J.; Sinto, R.; Subekti, D.; Adiwinata, R.; Waslia, L.; Loho, T.; Safari, D.; Widodo, D. Screening of methicillin-resistant Staphylococcus aureus nasal colonization among elective surgery patients in referral hospital in Indonesia. BMC research notes 2018, 11, 56.

8.        Price, C.S.; Williams, A.; Philips, G.; Dayton, M.; Smith, W.; Morgan, S. Staphylococcus aureus nasal colonization in preoperative orthopaedic outpatients. Clinical orthopaedics and related research 2008, 466, 2842-2847.

9.        Kim, D.H.; Spencer, M.; Davidson, S.M.; Li, L.; Shaw, J.D.; Gulczynski, D.; Hunter, D.J.; Martha, J.F.; Miley, G.B.; Parazin, S.J.; et al. Institutional prescreening for detection and eradication of methicillin-resistant Staphylococcus aureus in patients undergoing elective orthopaedic surgery. The Journal of bone and joint surgery. American volume 2010, 92, 1820-1826.

10.      Neidhart, S.; Zaatreh, S.; Klinder, A.; Redanz, S.; Spitzmuller, R.; Holtfreter, S.; Warnke, P.; Alozie, A.; Henck, V.; Gohler, A.; et al. Predictors of colonization with Staphylococcus species among patients scheduled for cardiac and orthopedic interventions at tertiary care hospitals in north-eastern Germany-a prevalence screening study. European journal of clinical microbiology & infectious diseases : official publication of the European Society of Clinical Microbiology 2018, 37, 633-641.

11.      Kobayashi, K.; Ando, K.; Ito, K.; Tsushima, M.; Morozumi, M.; Tanaka, S.; Machino, M.; Ota, K.; Ishiguro, N.; Imagama, S. Prediction of surgical site infection in spine surgery from tests of nasal MRSA colonization and drain tip culture. European journal of orthopaedic surgery & traumatology : orthopedie traumatologie 2018, 28, 1053-1057.

Reviewer 2 Report

 Searching for "orthopedic surgery MRSA nasal colonization" in Pubmed also found 33 articles. With so many papers, it can not be said that this area is only understood in a limited way. As this is the subject of a paper, the reason why the author states that it is limited should be explained in detail in comparison with the Pubmed publication. If the author can not properly explain this, they can not trust the specific research results they have issued.

Since this study is a study in a single region of Korea, it should be limited to Korea and the results can not be generalized globally.

The statistical analysis methods in Tables 2, 3 and 4 do not seem to be performed correctly. Are cigarettes and alcohol really not a risk factor in statistical analysis? Why do we need to assess the educational background of high school students or higher as a risk factor?

The authors' description of the correlation between obesity and MRSA carriage is missing.

If this study was conducted in Korea, you should describe not only guidelines for carrying MRSA in the US, but also guidelines for carrying MRSA in Korea.

Author Response

Response to Reviewer 2 Comments

Point 1: Searching for "orthopedic surgery MRSA nasal colonization" in PubMed also found 33 articles. With so many papers, it can not be said that this area is only understood in a limited way. As this is the subject of a paper, the reason why the author states that it is limited should be explained in detail in comparison with the PubMed publication. If the author can not properly explain this, they can not trust the specific research results they have issued.

Response 1: Thank you for your insightful advice. We have presented the references in the above sentence (line 59-60). We typed “Orthopedic surgery MRSA nasal colonization” in PubMed and reviewed all 33 articles as well as 1 related article (See the references located at the end of this response letter). We found that not all articles focused on the prevalence and risk factors of MRSA nasal colonization in orthopedic patients. Some articles focused on the decolonization of MRSA [1,2] and others were molecular studies [3,4]. Some of the articles focused on MRSA colonization among orthopedic surgeons rather than patients [5,6]. Only 5 articles satisfied the criteria (Prevalence and risk factors of orthopedic patients) [7-11]. Moreover, the number of patients included in our study (1577 patients) turned out to be relatively large compared with the number in these studies. We suggest that studies on the prevalence and risk factors of MRSA nasal colonization among orthopedic patients are limited; therefore, our results are valuable. (We have added additional files regarding the description of the searched articles.)

Point 2: Since this study is a study in a single region of Korea, it should be limited to Korea and the results can not be generalized globally.

Response 2: Thank you for your detailed feedback. We agree with your opinion that this study could be limited to a restricted region in Korea, and thus, the results cannot be generalized. Therefore, we added the word “in Korea” in the title (line 4).

Point 3: The statistical analysis methods in Tables 2, 3 and 4 do not seem to be performed correctly. Are cigarettes and alcohol really not a risk factor in statistical analysis?

Response 3: We apologize for the mistake. We found some errors regarding statistical methods and have revised the Tables 2, 3, and 4.

Point 4: Why do we need to assess the educational background of high school students or higher as a risk factor?

Response 4:  Thank you for your comment. In several studies, lower socio-economic status was found to be a risk factor for MRSA infection. We have added suitable references for this (line 107).

Point 5: The authors' description of the correlation between obesity and MRSA carriage is missing.

Response 5: Thank you for your insightful advice. We added the description of the correlation between obesity and MRSA carriage (line 206-208).

Pont 6: If this study was conducted in Korea, you should describe not only guidelines for carrying MRSA in the US, but also guidelines for carrying MRSA in Korea.

Response 6: Thank you for your comment. We added the guidelines for MRSA screening in Korea (line 50-54).

Reference

1.        Bajolet, O.; Toussaint, E.; Diallo, S.; Vernet-Garnier, V.; Dehoux, E. [Is it possible to detect Staphylococcus aureus colonization or bacteriuria before orthopedic surgery hospitalization?]. Pathologie-biologie 2010, 58, 127-130.

2.        Chen, A.F.; Heyl, A.E.; Xu, P.Z.; Rao, N.; Klatt, B.A. Preoperative decolonization effective at reducing staphylococcal colonization in total joint arthroplasty patients. The Journal of arthroplasty 2013, 28, 18-20.

3.        Trouillet-Assant, S.; Valour, F.; Mouton, W.; Martins-Simoes, P.; Lustig, S.; Laurent, F.; Ferry, T. Methicillin-susceptible strains responsible for postoperative orthopedic infection are not selected by the use of cefazolin in prophylaxis. Diagnostic microbiology and infectious disease 2016, 84, 266-267.

4.        Post, V.; Harris, L.G.; Morgenstern, M.; Geoff Richards, R.; Sheppard, S.K.; Fintan Moriarty, T. Characterization of nasal methicillin-resistant Staphylococcus aureus isolated from international human and veterinary surgeons. Journal of medical microbiology 2017, 66, 360-370.

5.        Schwarzkopf, R.; Takemoto, R.C.; Immerman, I.; Slover, J.D.; Bosco, J.A. Prevalence of Staphylococcus aureus colonization in orthopaedic surgeons and their patients: a prospective cohort controlled study. The Journal of bone and joint surgery. American volume 2010, 92, 1815-1819.

6.        Morgenstern, M.; Erichsen, C.; Hackl, S.; Mily, J.; Militz, M.; Friederichs, J.; Hungerer, S.; Buhren, V.; Moriarty, T.F.; Post, V.; et al. Antibiotic Resistance of Commensal Staphylococcus aureus and Coagulase-Negative Staphylococci in an International Cohort of Surgeons: A Prospective Point-Prevalence Study. PLoS One 2016, 11, e0148437.

7.        Nelwan, E.J.; Sinto, R.; Subekti, D.; Adiwinata, R.; Waslia, L.; Loho, T.; Safari, D.; Widodo, D. Screening of methicillin-resistant Staphylococcus aureus nasal colonization among elective surgery patients in referral hospital in Indonesia. BMC research notes 2018, 11, 56.

8.        Price, C.S.; Williams, A.; Philips, G.; Dayton, M.; Smith, W.; Morgan, S. Staphylococcus aureus nasal colonization in preoperative orthopaedic outpatients. Clinical orthopaedics and related research 2008, 466, 2842-2847.

9.        Kim, D.H.; Spencer, M.; Davidson, S.M.; Li, L.; Shaw, J.D.; Gulczynski, D.; Hunter, D.J.; Martha, J.F.; Miley, G.B.; Parazin, S.J.; et al. Institutional prescreening for detection and eradication of methicillin-resistant Staphylococcus aureus in patients undergoing elective orthopaedic surgery. The Journal of bone and joint surgery. American volume 2010, 92, 1820-1826.

10.      Neidhart, S.; Zaatreh, S.; Klinder, A.; Redanz, S.; Spitzmuller, R.; Holtfreter, S.; Warnke, P.; Alozie, A.; Henck, V.; Gohler, A.; et al. Predictors of colonization with Staphylococcus species among patients scheduled for cardiac and orthopedic interventions at tertiary care hospitals in north-eastern Germany-a prevalence screening study. European journal of clinical microbiology & infectious diseases : official publication of the European Society of Clinical Microbiology 2018, 37, 633-641.

11.      Kobayashi, K.; Ando, K.; Ito, K.; Tsushima, M.; Morozumi, M.; Tanaka, S.; Machino, M.; Ota, K.; Ishiguro, N.; Imagama, S. Prediction of surgical site infection in spine surgery from tests of nasal MRSA colonization and drain tip culture. European journal of orthopaedic surgery & traumatology : orthopedie traumatologie 2018, 28, 1053-1057.

Reviewer 3 Report

Abstract: Fine, but study design is missing.

Introduction: Fine

Material and Methods: Fine

Line 60 “ability to understand the scope and significance of the study” it seems to be a generic concept and too difficult to understand.

Line 87 “ICD-10-CM” is an unexplained acronym

Results: Line 109-110: the total is 100,2. Please correct

Line 112: “114 patientes” it is not clear how many of these 114 are those found positive in the outpatient department and after admission.

Table II and Table III: some totals are 100,2 or 100,1. Please correct

Remove 3D from figure 1

Discussion: Fine, but line 213 I would insert reference.

Environmental hygiene may play a possible role in spreading of MRSA. Hand washing is for sure a method for preventing spreading of germs- In addition it has been demonstrated that some objects, such as stethoscopes, are as contaminated as the hands. Is it possible a cross contamination during the hospital admission of these investigated patients? Authors could expand this part in the discussion. I suggest the following paper in case it can be useful.

-           “Tanning the bugs - a pilot study of an innovative approach to stethoscope disinfection” Messina G, Rosadini D, Burgassi S, Messina D, Nante N, Tani M, Cevenini G. J Hosp Infect. 2017 Feb;95(2):228-230. doi: 10.1016/j.jhin.2016.12.005.

Conclusions: Fine but minor revision required.

References: The bibliography could be updated

Author Response

Response to Reviewer 3 Comments

Point 1 :

Abstract: Fine, but study design is missing.

Response 1: Thank you for your comment. We added the study design in the abstract section. (Line 25-26)

Introduction: Fine

Material and Methods: Fine

Point 2:

Line 60 “ability to understand the scope and significance of the study” it seems to be a generic concept and too difficult to understand.

Response 2: Thank you for your comment. We also thought that such sentence would be quite awkward and deleted it. (Line 76-77 )

Point 3:

Line 87 “ICD-10-CM” is an unexplained acronym

Response 3: Thank you for your comment. We explained the acronym in the bracket. (Line 105-106)

Point 4:

Results: Line 109-110: the total is 100,2. Please correct                    

Response 4: Sorry for the mistake. We corrected the number. (Line 128-130)

Point 5:

Line 112: “114 patientes” it is not clear how many of these 114 are those found positive in the outpatient department and after admission.

Response 5: Thank you for comment. Of the 114 colonized samples, 99 (86.8%) samples had been performed after admission and 15 (13.2%) samples had been performed in the outpatient department. We added the results in the manuscript. (Line 132-134)

Point 6:

Table II and Table III: some totals are 100,2 or 100,1. Please correct

Response 6: Sorry for the mistake. We corrected the number.

Point 7:

Remove 3D from figure 1

Response 7: Thank you for your comment. We thought figure 1 would be inappropriate in the context and removed it.

Point 8:

Discussion: Fine, but line 213 I would insert reference.

Environmental hygiene may play a possible role in spreading of MRSA. Hand washing is for sure a method for preventing spreading of germs- In addition it has been demonstrated that some objects, such as stethoscopes, are as contaminated as the hands. Is it possible a cross contamination during the hospital admission of these investigated patients? Authors could expand this part in the discussion. I suggest the following paper in case it can be useful.

-           “Tanning the bugs - a pilot study of an innovative approach to stethoscope disinfection” Messina G, Rosadini D, Burgassi S, Messina D, Nante N, Tani M, Cevenini G. J Hosp Infect. 2017 Feb;95(2):228-230. doi: 10.1016/j.jhin.2016.12.005.

Response 8: Thank you for your comment. We added the reference you suggested on the manuscript (Line 239)

Point 9:

Conclusions: Fine but minor revision required.

Response 9: Thank you for your comment. We added our opinion on the conclusion.(Line 272-273)

Point 10:

References: The bibliography could be updated

Response 10 : Thank you for your comment. We added some other references through the revision.

Round  2

Reviewer 1 Report

The authors properly addressed all the issues in the current revision. The methodology is now clearly presented, the results and discussions are improved.

Reviewer 2 Report

From the authors' answers, I have no part to make additional comments.